# Synergistic Larvicidal and Pupicidal Effects of Monoterpene Mixtures Against *Aedes aegypti* with Low Toxicity to Guppies and Honeybees

**DOI:** 10.3390/insects16070738

**Published:** 2025-07-18

**Authors:** Sirawut Sittichok, Hataichanok Passara, Tanapoom Moungthipmalai, Jirisuda Sinthusiri, Kouhei Murata, Mayura Soonwera

**Affiliations:** 1School of Agriculture and Cooperatives, Sukhothai Thammathirat Open University, Nonthaburi 11120, Thailand; sirawut.sit@stou.ac.th; 2Office of Administrative Interdisciplinary Program on Agricultural Technology, School of Agricultural Technology, King Mongkut’s Institute of Technology Ladkrabang, Ladkrabang, Bangkok 10520, Thailand; hataichanok.pa@kmitl.ac.th; 3Department of Plant Production Technology, School of Agricultural Technology, King Mongkut’s Institute of Technology Ladkrabang, Ladkrabang, Bangkok 10520, Thailand; 64604012@kmitl.ac.th; 4Community Public Health Program, Faculty of Public and Environmental Health, Huachiew Chalermprakiet University, Samut Prakan 10540, Thailand; jiri_ja@yahoo.com; 5School of Agriculture, Tokai University, Kumamoto 862-8652, Japan; kmurata@agri.u-tokai.ac.jp

**Keywords:** *Aedes aegypti*, eucalyptol, *trans*-anethole, geranial, synergistic effect, non-target species, insecticides

## Abstract

One of the primary global public health issues is the vector *Aedes aegypti* L., which spreads a variety of arboviruses, especially dengue. Epidemic outbreaks of arboviruses occur frequently despite significant investments in programs to battle the vector because insects develop resistance to conventional insecticides and multiply. Thus, organic insecticides—monoterpenes—are regarded as one of the most important agents for containing this virus vector. The efficiency of pure monoterpenes and their mixtures—geranial, *trans*-anethole, *trans*-cinnamaldehyde, and eucalyptol—against larvae and pupae was investigated. The insecticidal efficacy of these oils, at concentrations of up to 400 µg/mL, was compared to that of temephos. Several mixtures were more effective than pure monoterpenes and temephos (an organophosphate larvicide). All formulations were harmless to mosquito-eating fish (guppies) and pollinators (honeybees). Our findings contribute new information on plant-derived mosquito eradication and underline the need to scientifically assess the current efficacy of anti-mosquito products.

## 1. Introduction

The *Aedes aegypti* (Linnaeus) mosquito is responsible for spreading the pathogens that cause dengue, chikungunya, Zika, and yellow fever. It is also notorious for its painful and persistent bites [1,2]. These diseases have major health effects, including irreversible harm to newborn brain development (microcephaly) and billions of dollars in costs to local economies [3,4]. The World Health Organization (WHO) has identified this mosquito as a major global public health issue because it has adapted effectively to cities and nations with large populations and hot, humid conditions [5,6].

Epidemic outbreaks of arboviruses occur frequently despite significant investments in programs to battle the vector because insect resistance mechanisms have increased the number of mosquitoes resistant to conventional insecticides [7,8,9].

Using synthetic insecticides is the most common method of mosquito control; however, they may negatively affect non-target organisms and lead to ecological imbalances [10]. Additionally, some insecticides are associated with health and environmental risks [11,12]. Temephos is a widely used chemical for controlling mosquito larvae [13], but several studies have reported its harmful effects on humans, e.g., eye irritation, acute toxicity, and reproductive effects. It acts as a neurotoxicant, inducing cholinesterase (ChE) inhibition, which includes hypoactivity, labored breathing, dry coughs, salivation, and muscle spasms or tremors [11,14]. Accordingly, there is an interest in replacing synthetic compounds with natural chemical compounds (e.g., essential oils (EOs) and monoterpenes, i.e., eucalyptol, geranial, *trans*-anethole, and *trans*-cinnamaldehyde) that can affect every stage of the vector’s life cycle, especially the larval and pupal stages, to reduce the growing number of infections [15,16]. These monoterpenes are not only eco-friendly, but they are also safe for humans and other aquatic species [15,16,17].

Many plant EOs have monoterpenes as their primary ingredients, which give plants their odor due to their relatively high vapor pressures [10]. A wide range of biological activities have been exhibited by monoterpenes. They are routinely employed in food chemistry, chemical ecology, and the pharmaceutical industry, including as insecticides and insect repellents [10,18,19]. Several researchers have reported that pure and mixed monoterpenes are highly toxic to mosquitoes, especially eucalyptol, geranial, *trans*-anethole, and *trans*-cinnamaldehyde (Table 1). Their chemical structures were reported by Sigma-Aldrich (Figure 1). However, due to their high volatility and lipophilic nature, monoterpenes may have limited effectiveness against mosquitoes in the natural environment [20].

Furthermore, several monoterpenes have been found to be non-toxic to other tested species (yellow mealworm beetle: *Tenebrio molitor*; planktonic crustacean: *Daphnia magna*; earthworm: *Eisenia fetida*; western mosquitofish: *Gambusia affinis*; molly: *Poecilia latipinna*; zebrafish (*Danio rerio*); and stingless bee: *Tetragonula pagdeni*) [17,19,29,30,31,32]. On the other hand, temephos has shown highly dangerous side effects on non-target organisms [17,32].

This study assessed the efficacy of pure and combined formulations of monoterpenes, namely, eucalyptol, geranial, *trans*-anethole, and *trans*-cinnamaldehyde, against *Ae*. *aegypti* larvae and pupae. Since these monoterpenes showed very low lethal doses (LD_50_) and lethal concentrations (LC_50_), they were deemed safe for both people and the environment [33,34,35,36,37], in addition to demonstrating insecticidal and other pharmacological actions [22,35,38,39,40]. All mixture treatments were selected based on our previous reports [17,32]. The synergistic effects of the mixtures and their biosafety were evaluated against non-target species: honeybees (*Apis mellifera*) and guppies (*Poecilia reticulata* Peters). Both non-target species are common predators and pollinators in tropical areas such as Thailand and Southeast Asia [32,41]. In addition, all formulations showed morphological damage in mosquito larvae and pupae, observed using optical microscopy. The findings from this study provide useful data for the further development of EO-derived formulations for eradicating mosquito larvae and pupae.

## 2. Materials and Methods

### 2.1. Mosquitoes

*Ae*. *aegypti* mosquitoes originated from the Department of Entomology at the Armed Forces Research Institute of Medical Sciences (AFRIMS), Ratchatewi, Bangkok, Thailand. Since 2020, these mosquitoes have been cultivated in our laboratory at the School of Agricultural Technology, KMITL, under controlled conditions with a temperature range of 26 ± 1 °C, humidity levels at 60 ± 3%, and 12 h light and dark periods. They were also free from pathogens and insecticides as per WHO guidelines [17,42]. Fish food pellets (from Sakura^®®^ Gold, which has a high protein content of 35%; TSDP (Thailand) Co., Ltd., Samut Sakhon, Thailand) were used to feed the mosquito larvae. The fourth larval stage (4–5 mm in length) and 2-day-old pupae were subjected to larvicidal and pupicidal tests.

### 2.2. Chemicals and Treatment Formulations

Eucalyptol 99% (CAS-No: 470-82-6, extracted from eucalyptus oil), geranial 96% (CAS-No: 5392-40-5, from lemongrass oil), *trans*-anethole 99% (CAS-No: 4180-23-8, from star anise oil), and *trans*-cinnamaldehyde 98% (CAS-No: 104-55-2, from cinnamon oil) were purchased from Sigma-Aldrich Company Ltd. (Saint Louis, MO, USA). Stock solutions were produced using 70% (*v*/*v*) ethanol (T.S. Interlab Company Ltd., Bangkok, Thailand) and stored in sealed brown bottles. Stock solutions of pure forms of eucalyptol, geranial, *trans*-anethole, and *trans*-cinnamaldehyde were used as the 1% concentrations; binary mixtures were eucalyptol + *trans*-cinnamaldehyde, eucalyptol + geranial, eucalyptol + *trans*-anethole, and *trans*-anethole + geranial at 1:1 ratios. World Health Organization recommends 1% (*w*/*w*) of temephos (Sai GPO 1^®^, The Government Pharmaceutical Organization, Pathumthani, Thailand) to be effective against mosquito larvae [43].

The criteria for characterizing the efficiency of EO constituents as larvicides and pupicides varied from several sources [11]. In this study, we used the criteria established in previous reports [16,17], which set 200 µg/mL as the minimum, based on the median lethal concentration (LC_50_). Therefore, EO constituents with LC_50_ ≥ 200 µg/mL were classified as less active, whereas those with LC_50_ ≤ 200 µg/mL were classified as highly active.

### 2.3. Toxicity Bioassay

The standard WHO procedure [42] was slightly modified to investigate the toxicity against *Ae*. *aegypti* larvae and pupae. Two test concentrations of each monoterpene were prepared (200 and 400 µg/mL). In this experiment, the larvae were not fed with any nourishment. Then, 100 mL of distilled water was placed in a beaker with ten 4th instar larvae or pupae. Temephos and distilled water were used as positive and negative controls, following WHO recommendations. The assay was performed five times for each treatment. The clearest sign that the larvae and pupae were dead was their inability to surface and disrupt the water or their failure to dive into the water. Larval mortality was recorded at 5, 10, 15, and 30 min, and 1, 2, and 24 h, while pupal mortality was recorded at 5, 10, 15, and 30 min, and 1, 2, 24, 48, and 72 h. Mortality rates of larvae and pupae (%MT) were computed using the following equation [16,17]:Mortality rate (%MT) = DM/TM × 100(1)
where DM represents all of the dead larvae or pupae and TM represents all of the treated larvae or pupae.

In the following formulae, factors were determined at 24 h for larvae or at 72 h for pupae. The mortality index (MI) was determined as follows [17]:MI = %MT_treat_/%MT_temephos_(2)
where %MT_treat_ is the %mortality of the tested formulations and %MT_temephos_ is the %mortality of 1% temephos.

MI less than 1 means that the treatment was more harmful to the larvae or pupae than temephos. MI is relative toxicity.

Binary mixtures were more effective than pure formulations, as indicated by the increased mortality value (IMV). IMV was calculated as follows [41]:IMV = [%MT_mix_ − (sum %MT_pure_)/%MI_mix_] × 100(3)
where %MT_mix_ is the %mortality of the binary mixture and %MT_sing_ is the %mortality of the pure formulations.

Binary mixtures were more effective than pure formulations at the same concentration, indicated by the synergistic mortality index (SRI). SRI was calculated using the following equation [16,17]:SRI = LT_50 mix_/(LT_50 pure EO 1_ + LT_50 pure EO 2_)(4)
where LT_50 mix_ is the LT_50_ of the binary mixture and LT_50 pure EO 1 or 2_ is the LT_50_ of the pure formulations.

The relative synergy is indicated by SRI: if SRI is less than 1, a synergistic impact is present; if SRI is more than 1, no synergy is present.

The higher larvicidal and pupicidal activity of binary mixtures compared to pure formulation is known as increased concentration value (ICV). ICV was calculated as follows [16]:ICV = LC_50 mix_/(LC_50 pure EO 1_ + LC_50 pure EO 2_)(5)
where LC_50 mix_ is the LC_50_ of the binary mixture at 24 h for larvae or 72 h for pupae, and LC_50 pure EO 1 or 2_ is the LC_50_ of the pure formulations.

Relative toxicity is indicated by ICV: if ICV is less than or equal to 0.2, the treatment is seriously toxic to larvae and pupae.

### 2.4. Microscopic of Morphological Changes

Following the toxicity bioassay, the treated larvae and pupae’s morphological changes—both internal and external—were examined using a stereomicroscope (Nikon^®^ Type 102, Hollywood International Company Limited, Ratchathewi, Bangkok, Thailand), captured using an Olympus^®^ EP50 digital camera, Hollywood International Company Limited, Ratchathewi, Bangkok, Thailand at the Microscopy Centre, School of Agricultural Technology, KMITL [16], and categorized.

### 2.5. Toxicity Bioassay of Non-Target Species

#### 2.5.1. Bioassay of Guppies

Guppy predators were purchased from Molly Fish Farm Thailand, an organic farm located in Nakhon Pathom Province, Thailand (13.82934 °N, 100.10515 °E). Following the previous methods [32,44], this study evaluated the toxicity of all formulations against guppies. In a 400 × 600 × 300 mm plastic container, 100 fish were housed in 80 L of clean water at 32 ± 5 °C, 75 ± 5% RH, with 12 h light and 12 h dark periods. Fish food pellets (from Sakura^®^ Gold, with a high protein content of 35%; TSDP (Thailand) Co., Ltd., Samut Sakhon, Thailand) were used to feed the guppies. Only males were employed in this bioassay because they were more readily available and in higher demand in the market. Five liters of clean water was placed in a plastic container with ten adult male guppies (diameter: 350 mm; height: 180 mm). The concentrations of each treatment were 5000, 10,000, and 15,000 µg/mL [18]. The distilled water was used as a negative control. Mortality rate and abnormal behaviors in the guppies were recorded for 5 days post-treatment. Mortality rates (%MR) were computed as follows [17,19,32]:Mortality rate (%MT) = DG/DT × 100(6)
where DG is the number of dead guppies and DT is the number of treated guppies.

#### 2.5.2. Bioassay of Honeybees

Honeybee workers were collected from an organic sugar cane farm (Bang Sao Thong District, Samut Prakan Province, Thailand, 13.72058° N, 100.76511° E). The toxicity of the tested formulations was tested using these bees [19]. A total of 150 worker bee pollinators were transferred into an insect cage (350 × 350 × 350 mm) and transported to the entomological laboratory within 1 h of collection. The collected bees were maintained at 26 ± 1 °C, 75 ± 5% RH, and provided with a 50% sucrose solution until the start of the bioassay. To begin the experiment, the topical test consisted of applying 1 µL of each tested formulation to the dorsal part of each tested bee. Distilled water was used as a negative control. After that, ten pollinators were transferred into a plastic box (120 × 170 × 115 mm) and fed a sugar solution. Bee mortality was recorded at 24 h after treatment. Mortality rates (%MR) were computed using Formula (6).

The biosafety index (BI) was determined as follows [19,28]:BI = %MT_non-target_/%MT_target_(7)
where %MT_non-target_ is the %mortality of the non-target species (honeybees and guppies) and %MT_target_ is the %mortality of the target species (mosquito larvae and pupae) at 400 µg/mL treatment concentration.

The tested formulation was toxic to the non-target species if its BI was greater than 0.90, whereas it was benign if its BI was less than 0.90.

### 2.6. Statistical Analysis

A totally randomized design was employed in all bioassays, and five replications of each treatment were conducted. Tukey’s test was utilized to verify differences across several treatment groups, and one-way ANOVA was utilized to evaluate the mean mortality for the larvicidal and pupicidal assays and the mean mortality ± standard error (S.E.) for the non-target bioassays [45]. For all statistical tests, the standard *p* < 0.05 threshold was applied. A probit analysis of mortality (the number of larvae and pupae that had died at 24 and 72 h after exposure) was used to calculate the time it took for a substance to reach 50% mortality (LT_50_) and the concentration that caused 50% mortality (LC_50_) against the larvae and pupae. A generalized linear model with a binomial distribution was used in simple regression to evaluate the larvicidal and pupicidal efficacy against *Ae*. *aegypti* [46]. A correlation coefficient, R^2^, indicated acceptable linearity. All the experimental analyses used IBM’s SPSS version 28 (Armonk, NY, USA) software package.

## 3. Results

### 3.1. Toxicity of Pure and Mixtures of Monoterpenes Against Ae. aegypti Larvae and Pupae

Figure 2 shows the larvicidal activity of both pure monoterpenes and their mixtures against *Ae. aegypti*, presented as a regression of toxicity to larvae over time. Several regression lines had an R^2^ value close to 1. After 24 h, all pure monoterpenes were significantly less effective than the binary mixtures. Higher concentrations were notably more effective than lower ones. Among the pure monoterpenes, *trans*-anethole at 400 µg/mL had the highest mortality rate at 98%, which was significantly different from temephos (100% mortality). The highest mortality rate among the mixtures was 100%, achieved by both eucalyptol + geranial and eucalyptol + *trans*-anethole at 400 µg/mL, with no significant difference compared to temephos. Distilled water, used as the negative control, had no effect on the larvae.

In terms of MI, *trans*-anethole and *trans*-cinnamaldehyde at 400 µg/mL, as well as all mixtures at 400 µg/mL, were equally effective as temephos, with an MI of 1. Pure monoterpenes and mixtures at 200 µg/mL were less effective than temephos, with MI values ranging from 0.03 to 0.8. The lowest larvicidal activity was observed with *trans*-anethole at 200 µg/mL.

The regression toxicity results of *Ae*. *aegypti* pupae versus exposure time for pure compounds and mixtures are shown in Figure 3. All pure monoterpene compounds were significantly less effective than any mixture after 72 h of exposure. Mixtures were significantly more effective against *Ae*. *aegypti* pupae at 400 µg/mL than at lower concentrations. The highest mortality among the pure monoterpenes was 39%, achieved by geranial 400 µg/mL, but this was significantly lower than that of temephos (90% mortality). However, among the mixtures, the highest mortality of 100% was achieved by *trans*-anethole + eucalyptol and *trans*-anethole + geranial at 400 µg/mL, i.e., comparable in efficacy to temephos. In addition, the distilled water negative control had no effect on pupae. For MI, the strongest activity was shown by eucalyptol + *trans*-anethole and *trans*-anethole + geranial. This MI was 1.1 times higher than that of temephos. Other mixtures were less effective than this temephos concentration, with 0.02 ≤ MI ≤ 0.7. The lowest larvicidal activity was provided by *trans*-anethole alone at 200 µg/mL.

The toxicity of larvicidal and pupicidal activities against *Ae*. *aegypti* was estimated using the LT_50_ value, as shown in Figure 4. All mixtures showed stronger larvicidal and pupicidal activities against *Ae*. *aegypti* than pure monoterpenes. Among the pure monoterpenes, *trans*-anethole and geranial at 400 µg/mL showed the highest larvicidal and pupicidal activities with LT_50_ of 8 and 75 h. Particularly, the mixture of *trans*-anethole + geranial at 400 µg/mL exhibited the strongest larvicidal and pupicidal activities with LT_50_s as short as 0.2 (larvae) and 1.8 (pupae) h, and were much more effective than temephos (LT_50_ = 1.1 h for larvae and 48.3 h for pupae).

In terms of the LC_50_ value, all mixtures were more toxic to *Ae*. *aegypti* larvae and pupae than pure monoterpenes (Table 2 and Table 3). The mixture of eucalyptol + *trans*-anethole was even more effective against larvae than other mixtures with LC_50,90_ = 176 and 228 µg/mL, respectively, while the mixture of *trans*-anethole + geranial provided LC_50,90_ of 167 and 217 µg/mL, respectively, for pupae. Based on the lowest ICV, the strongest larvicidal and pupicidal activities were exhibited by *trans*-anethole + geranial.

The IMVs of pure monoterpenes and mixtures against both larvae and pupae of *Ae*. *aegypti* are shown in Figure 5. All mixture formulations improved the IMV by 25–95% for larvae and 51–92% for pupae compared to pure monoterpenes.

In addition, the mixtures were more effective against *Ae. aegypti* than the pure compounds, with an SMI ranging from 0.005 to 0.6. Three formulations—eucalyptol + geranial, eucalyptol + *trans*-anethole, and *trans*-anethole + geranial at 400 µg/mL—exhibited the highest synergistic effects on larvae and pupae, with an SI of 0.01, except the mixture of eucalyptol + geranial had a lower synergistic effect on pupae (see Figure 6).

### 3.2. Morphological Changes After Treatment with Monoterpenes

A light microscopy image shows changes in the external morphology of *Ae. aegypti* larvae—see Figure 7. Significant changes to the cuticles of the treated larvae (dorsal) were shown compared to the control group (Figure 7A,I). The damages were extensive: darkening of the cuticles, head and abdominal segments becoming misshapen, and destruction of the exoskeleton of the larvae, as well as the tracheal system becoming translucent and the anal papillae becoming swollen (Figure 7B–H). In particular, damage to or swelling of the respiratory siphon of the larvae was observed in all treatments (Figure 7J).

The treated pupae exhibited significant cell damage to the cuticles on the head, cephalothorax, and abdomen, compared to the control group (Figure 8A,I). Damages also included dark pigment cells in these areas (Figure 8B–H) and swollen breathing openings and respiratory trumpets (Figure 8J).

### 3.3. Efficacy on Non-Target Species

The mortality rates of adult guppies after 5 days of exposure are shown in Figure 9. Pure monoterpene and mixture formulations showed very low toxicity to the adults, with a mortality rate ranging from 2% to 14%. Pure eucalyptol and the mixture of eucalyptol + geranial showed low toxicity to the honeybees, with only 1% mortality. The mortality rate provided by other formulations ranged from 4% to 20% (Figure 9B). Distilled water, the negative control, also had no effect on guppies and honeybees.

Figure 10 shows that both pure monoterpene and the mixture formulations provided a low BI, from 0.01 to 0.87 < 0.9, signifying that all treatments were much safer for guppies and honeybees.

## 4. Discussion

Mosquito control is increasingly difficult due to its quick development of resistance to several insecticides, especially organophosphate and pyrethroid insecticides [47,48]. Usually, mosquito control is based on both larvicidal and adulticidal treatments, and the negative impact of insecticidal residues in aquatic and soil environments on non-target species, such as pollinators and aquatic and soil invertebrates, must be taken into account [49,50]. EOs and their monoterpene compounds represent the best option as organic alternatives to synthetic insecticides for mosquito control because they have been widely reported to possess remarkable insecticidal activity and to be safe for non-target species [51,52]. Moreover, the application of EO-based larvicides and pupicides to inhibit the development of the mosquito life cycle at the immature stages is a good strategy for controlling mosquito populations [15,16,17]. Our findings are significant in that two mixtures of *trans*-anethole + eucalyptol and *trans*-anethole + geranial showed high potential as alternative larvicides and pupicides against the immature stages of *Ae. aegypti*.

Several mixtures of monoterpenes showed a highly synergistic insecticidal activity compared to the corresponding individual components [53,54]. Importantly, the outcome of the synergy was the reduction in the amounts and hence the costs of the monoterpenes used when the mixture was applied for mosquito control [55,56,57]. In this study, all mixtures, as opposed to individual components, showed strong synergistic effects against *Ae. aegypti* larvae and pupae, with a higher mortality rate, shorter LT_50_ (h), higher IMV, lower SMI, and lower ICV, especially the mixtures of eucalyptol + *trans*-anethole and *trans*-anethole + geranial. These findings were consistent with other works that showed strong synergy. One study reports that a 1:1 combination of geranial + *trans*-cinnamaldehyde at 200 µg/mL showed strong larvicidal and pupicidal activities against *Ae. aegypti* [17]. Two mixtures, geranial + *trans*-cinnamaldehyde and D-limonene + *trans*-cinnamaldehyde, in a 1.5:1.5 ratio showed high ovicidal effects against the eggs of *Ae. aegypti* and *Ae. albopictus* [32]. The two combinations of 1,8-cineole + α-pinene and carvone + (R)-pulegone were strongly toxic to adults of *Culex pipiens* [56]. The two combinations of geranial + *trans*-anethole in 1:1 and 0.5:0.5 ratios showed strong repellent and adulticidal activities against housefly (*Musca domestica*) and mosquito adults [53,57]. Similarly, some mixtures of monoterpenes from 1,8-cineole + ϒ-terpinene, 1,8-cineole + p-cymene, 1,8-cineole + citronella, 1,8-cineole + linalool, 1,8-cineole + (R)-pulegone, and *trans*-anethole + eugenol showed high synergistic adulticidal effect against houseflies [58,59,60].

Temephos is an organophosphate larvicide that has been approved by the World Health Organization for the control of *Ae. aegypti* larvae [11,61]. After a long period of its use, the resistance of *Ae. aegypti* to temephos was found to be widespread in several countries [62,63]. Our findings of the MI showed that, currently, the mixtures of *trans*-anethole + eucalyptol and *trans*-anethole + geranial were even more potent than temephos as alternative larvicides and pupicides. Supported by the findings of several studies, it was found that 1% *w/w* temephos was less effective in terms of ovicidal, larvicidal, and pupicidal activities than some essential oils and their major monoterpenes, especially certain mixtures of them [16,17,32]. In Mexico and Brazil, *Ae. aegypti* larvae exhibited high resistance to temephos with a low mortality rate ranging from 4% to 62% and a high resistance ratio from 6.1 to 16.8 [63,64].

Several EOs and their monoterpenes have been found to cause morphological abnormalities in various stages of mosquito development. For example, lemongrass EO and clove (*Syzygium aromaticum*) EO led to deformed *Ae. aegypti* and *Anopheles dirus* larvae, with elongated thoraxes, abdomens lacking normal larval characteristics, and loss of terminal abdominal segments, siphon tubes, saddles, and hair tufts [65]. In this study, light microscopy images showed significant morphological impacts, especially on the respiratory siphons and trumpets of larvae and pupae of *Ae. aegypti* after treatment with both pure and mixed formulations. These findings are similar to other reports of significant external changes in *Ae. aegypti* and *Ae. albopictus* larvae and pupae after treatment with D-limonene and *trans*-anethole [16,17]. The changes were caused by a mixture of geranial + *trans*-cinnamaldehyde (1:1) and a combination of methyl cinnamate + linalool (1:4) [28]. Additionally, there were reports of significant external and internal changes in *Ae. aegypti* larvae induced by R-limonene, such as a larval disorder that reduces the total sugar level of the third-instar larvae and cytoplasmic vacuolization in the epithelial lamina of the midgut [66].

Several monoterpenes contained in EOs are good penetrants that increase their biological activity. They are neurotoxic to insect pests in multiple mechanisms of action; for instance, they show toxicity to octopamine synapses and gamma-aminobutyric acid (GABA), and inhibit acetylcholinesterase (AChE) enzymes [14,67]. The larvicidal and pupicidal mechanisms of action of eucalyptol, geranial and *trans*-anethole demonstrated their interaction with AChE enzymes, indicating that monoterpene strongly interacted with these enzymes [16,17,32,68]. Three monoterpenes are toxic to the cuticle, head, tracheal system, and respiratory siphons of *Ae. aegypti* larvae and the heads, cephalothorax, and respiratory trumpets of the pupae (see Figure 6 and Figure 7) [17]. They are toxic to mosquitoes by inhibiting AChE enzymes of neuroreceptors and nerve cells, and they induce paralysis and mortality in larvae and pupae [17,68]. In addition, knowing the mechanisms of action is important for researchers to improve the efficacy and quality of EO-based insecticides [14,68].

Even though monoterpenes showed high effectiveness against insect pests, they exhibited low toxicity to mammals, non-target vertebrates, invertebrates, and pollinators.

Moreover, their environmental persistence is short [69,70]. In this study, all pure components and mixtures were found to be relatively safe for guppies and honeybees, especially the mixtures of eucalyptol + *trans*-anethole and *trans*-anethole + geranial. The mortality rate was low (<20%), and BI was lower than 0.14. Supported by several works, a 1:1 mixture of geranial + *trans*-anethole was not toxic to guppies, mollies (*P. latipinna*), stingless bees (*T. pagdeni*), or dwarf honeybees (*Apis florea*) [16,19]. Additionally, 1:1 mixtures of geranial + *trans*-cinnamaldehyde and D-limonene + *trans*-cinnamaldehyde showed very low toxicity to guppies and mollies, with an LC_50_ of 4092 to 4554 ppm and BI lower than 1.5 [32]. In contrast, 1% temephos showed high toxicity with an LC_50_ of 299 to 527 ppm [32]. Similarly, temephos at a lower dose of 1 ppm was highly toxic to guppies, causing 100% mortality after 10 days of exposure [17]. At a dose of 10 mg/L, temephos caused leukocyte death in guppies, induced chronic effects on immune response cells, and decreased brain AChE activities [70,71]. Importantly, temephos and its oxidized derivatives have been reported to cause AChE inhibition in humans and mammals, along with other toxic side effects such as genotoxic effects, DNA fragmentation in blood cells, lymphocytes, headaches, and hyperactivity [72,73]. In contrast, mixtures of eucalyptol + *trans*-anethole and *trans*-anethole + geranial were not only harmless to guppies and honeybees but also safe for humans and used in both the food and drug industries [74,75]. *Trans*-anethole is used in the food industry as an aromatic and flavoring substance, and it is also used as medicine for managing neurological disorders [34]. Eucalyptol possesses pharmacological properties such as antioxidant and anti-inflammatory effects and is mostly used for respiratory and cardiovascular treatments [35]. Similarly, geranial has been used in traditional medicine in Asia for centuries [36,76].

Two mixtures in particular, eucalyptol + *trans*-anethole and *trans*-anethole + geranial, had highly synergistic larvicidal and pupicidal activities. They increased the mortality rate of *Ae. aegypti* larvae and pupae to more than 25–92%, and they are also safe for non-target species. Therefore, they should be used as a natural alternative insecticide for the control of *Ae. aegypti* to reduce the prevalence of dengue fever and other vector-borne diseases.

## 5. Conclusions

Our study showed that two mixture formulations at 400 µg/mL, eucalyptol + *trans*-anethole and *trans*-anethole + geranial, exerted synergistic actions and were more effective than the currently used temephos 1% (*w*/*w*). They should be further developed as an aqueous solution with low concentrations (160–180 µg/mL) to control immature-stage mosquitoes in their breeding sites, in households, and in other epidemic areas. The larvicidal and pupicidal efficiency of the two mixtures in the field and biosafety assays for humans and pets should be further studied.

## Figures and Tables

**Figure 1 insects-16-00738-f001:**
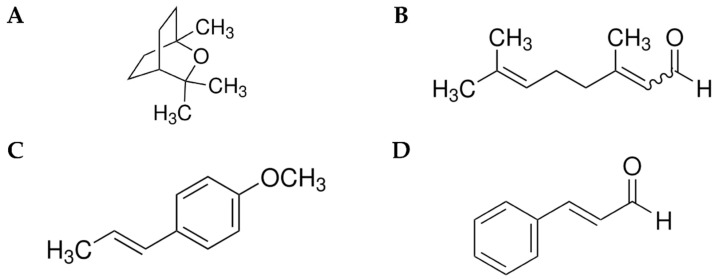
Chemical structure formulas from Sigma-Aldrich: eucalyptol (**A**), geranial (**B**), *trans*-anethol (**C**), and *trans*-cinnamaldehyde (**D**).

**Figure 2 insects-16-00738-f002:**
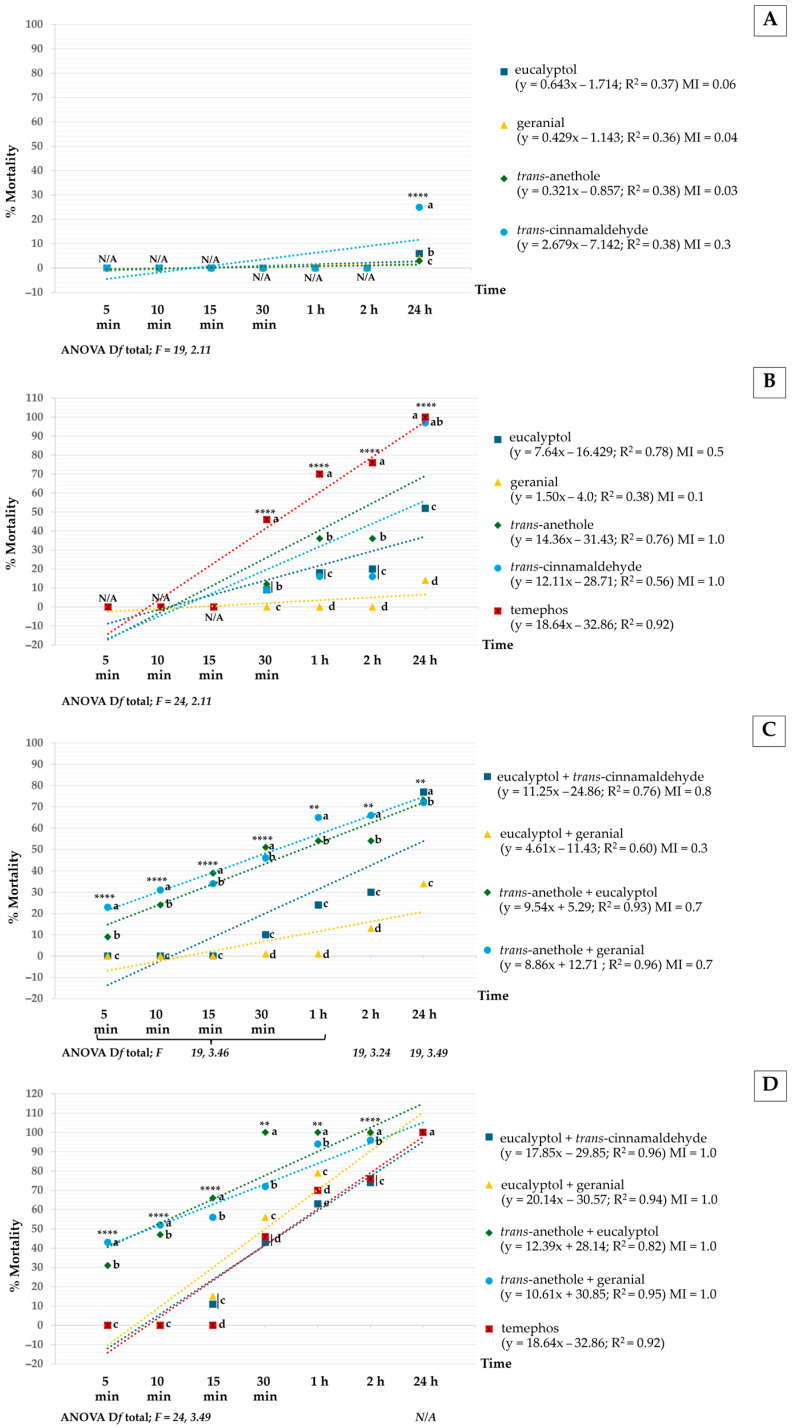
Mortality rate versus exposure time against *Ae*. *aegypti* larvae in all treatments: pure compounds at 200 µg/mL (**A**) and 400 µg/mL (**B**) and mixtures at 200 µg/mL (**C**) and 400 µg/mL (**D**). Note: Values that are accompanied by different letters (a–e show significant differences between the treatments. ** for *p* < 0.01; **** for *p* < 0.0001; N/A = not available.

**Figure 3 insects-16-00738-f003:**
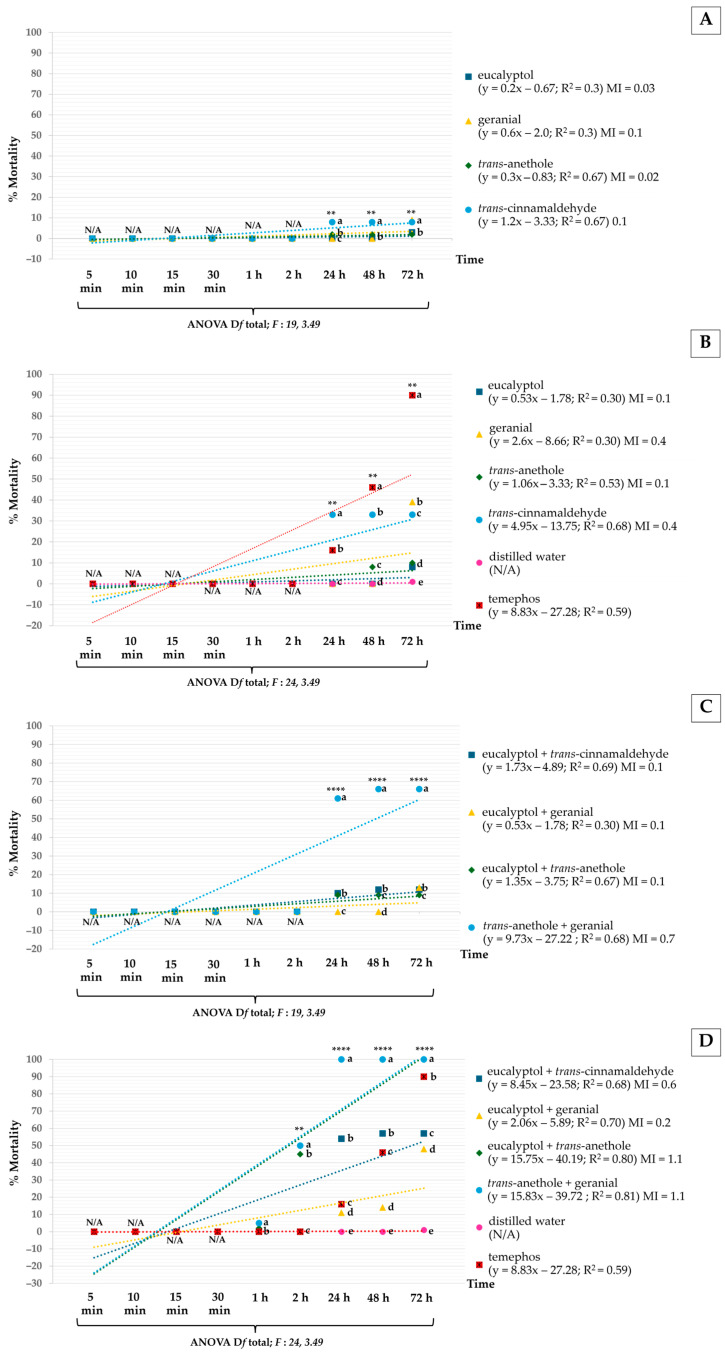
Percentage mortality versus exposure time for *Ae*. *aegypti* pupae under all tested formulations: pure monoterpene at 200 µg/mL (**A**) and 400 µg/mL (**B**) and mixture formulations at 200 µg/mL (**C**) and 400 µg/mL (**D**). Note: Values that are accompanied by different letters (a–e) show significant differences between the formulations. ** for *p* < 0.01; **** for *p* < 0.0001; N/A = not available.

**Figure 4 insects-16-00738-f004:**
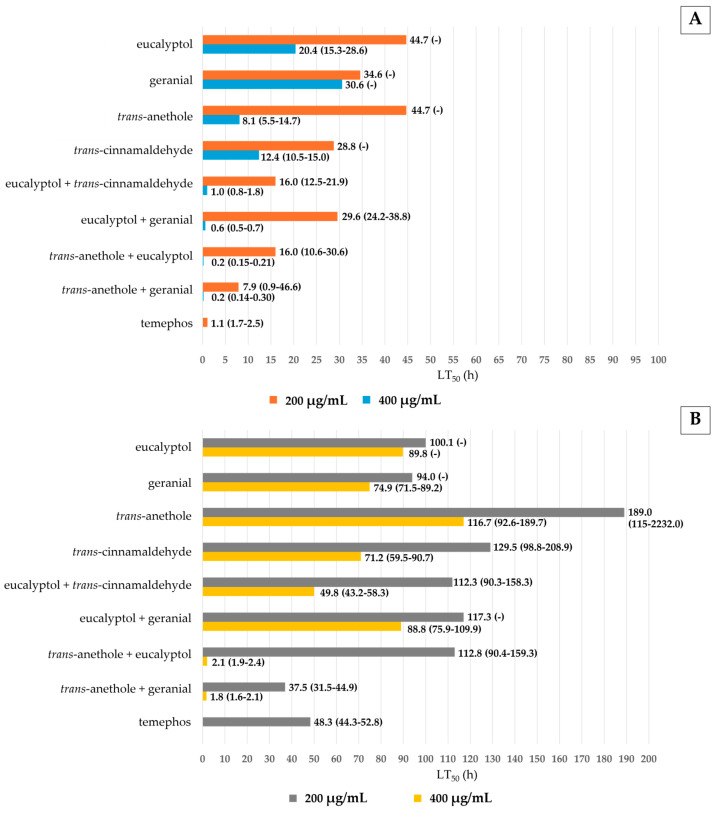
Lethal times (LT_50_) of pure monoterpenes and mixtures vs. temephos against *Ae*. *aegypti* larvae (**A**) and pupae (**B**). Note: LT_50_ = lethal time that kills 50% of the exposed organisms; *LL* 95% is the lower confidence limit and *UL* 95% is the upper confidence limit.

**Figure 5 insects-16-00738-f005:**
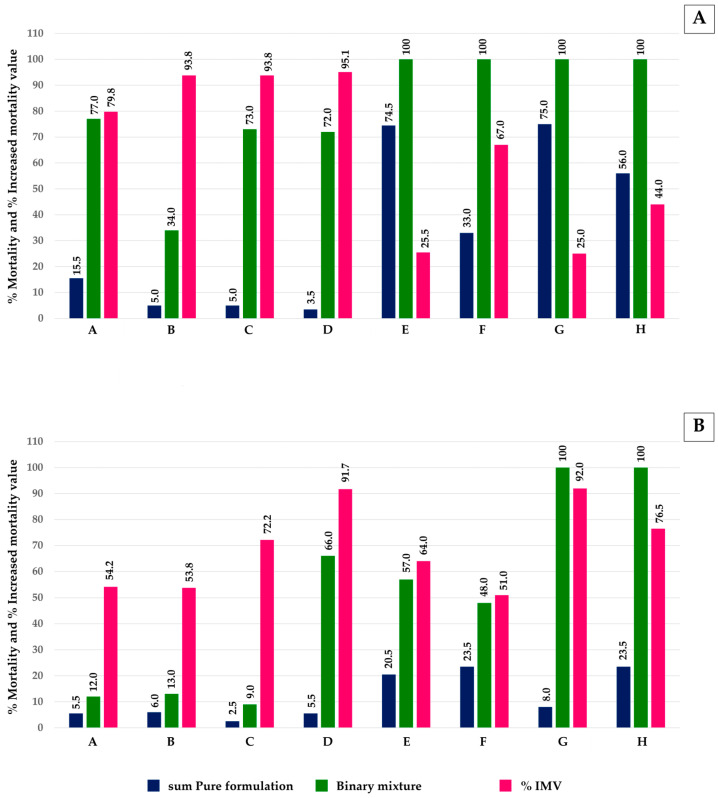
Increased mortality value (IMV) against *Ae*. *aegypti* of mixture formulations at 200 and 400 µg/mL versus corresponding pure monoterpenes: (**A**) larvae and (**B**) pupae. Note: **A**, eucalyptol + *trans*-cinnamaldehyde 200 µg/mL; **B**, eucalyptol + geranial 200 µg/mL; **C**, eucalyptol + *trans*-anethole 200 µg/mL; **D**, *trans*-anethole + geranial 200 µg/mL; **E**, eucalyptol + *trans*-cinnamaldehyde 400 µg/mL; **F**, eucalyptol + geranial 400 µg/mL; **G**, eucalyptol + *trans*-anethole 400 µg/mL; and **H**, *trans*-anethole + geranial 400 µg/mL.

**Figure 6 insects-16-00738-f006:**
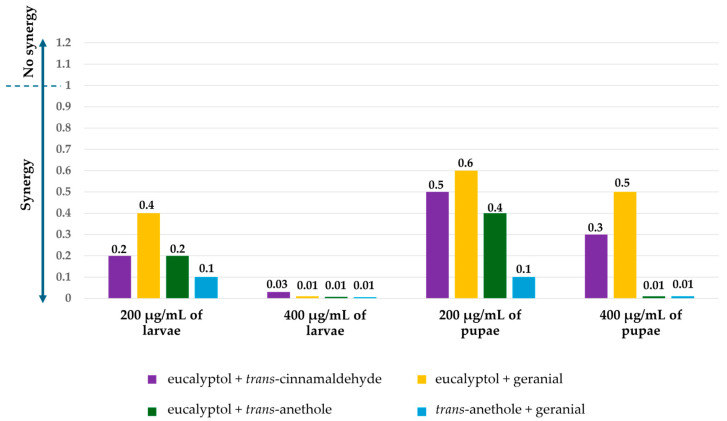
Synergistic mortality index (SMI) against *Ae*. *aegypti* larvae and pupae of several mixture formulations at 200 and 400 µg/mL.

**Figure 7 insects-16-00738-f007:**
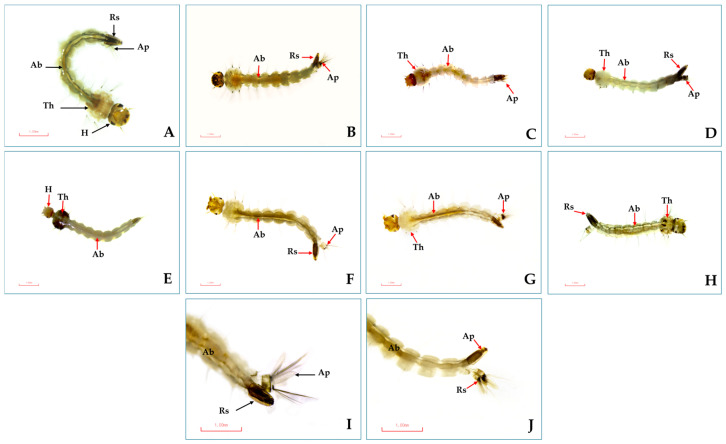
External morphology of *Ae*. *aegypti* larvae after 24 h of bioassay: (**A**) normal larvae surface, with head (H), thorax (Th), and abdomen (Ab), and (**I**) normal respiratory siphon (Rs) and anal papillae (Ap) (black arrow). Morphological changes showing the damage and swelling in abnormal head, abdomen, and thorax surfaces after the larvae were exposed to eucalyptol (**B**), geranial (**C**), *trans*-anethole (**D**), *trans*-cinnamaldehyde (**E**), eucalyptol + *trans*-anethole (**F**), and *trans*-anethole + geranial (**G**). Morphological damage by monoterpene formulations similar to temephos (**H**) (red arrow). In addition, all test formulations showed damage to the respiratory siphon and anal papillae (**J**) (red arrow).

**Figure 8 insects-16-00738-f008:**
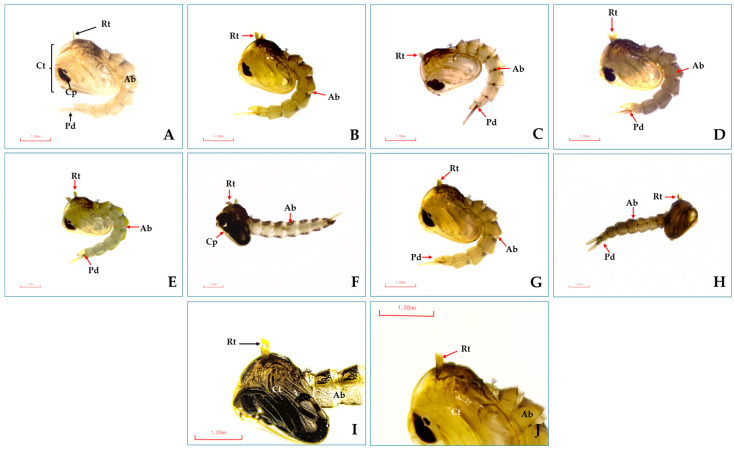
External morphology of *Ae*. *aegypti* pupae after 24 h of bioassay: (**A**) normal pupae surface with cephalothorax (Ct), abdomen (Ab), compound eye sheath (Cp), and paddies (Pd), and (**I**) normal respiratory trumpet (Rt) (black arrow). Morphological changes showing the abnormality of the head, cephalothorax, abdomen, and terminal abdominal structure surfaces after the pupae were exposed to eucalyptol (**B**), geranial (**C**), *trans*-anethole (**D**), *trans*-cinnamaldehyde (**E**), eucalyptol + *trans*-anethole (**F**), and *trans*-anethole + geranial (**G**). Morphological damages caused by monoterpene formulations similar to temephos (**H**) (red arrow). All test formulations showed swelling of the respiratory trumpet and related structures of the trumpet (**J**) (red arrow).

**Figure 9 insects-16-00738-f009:**
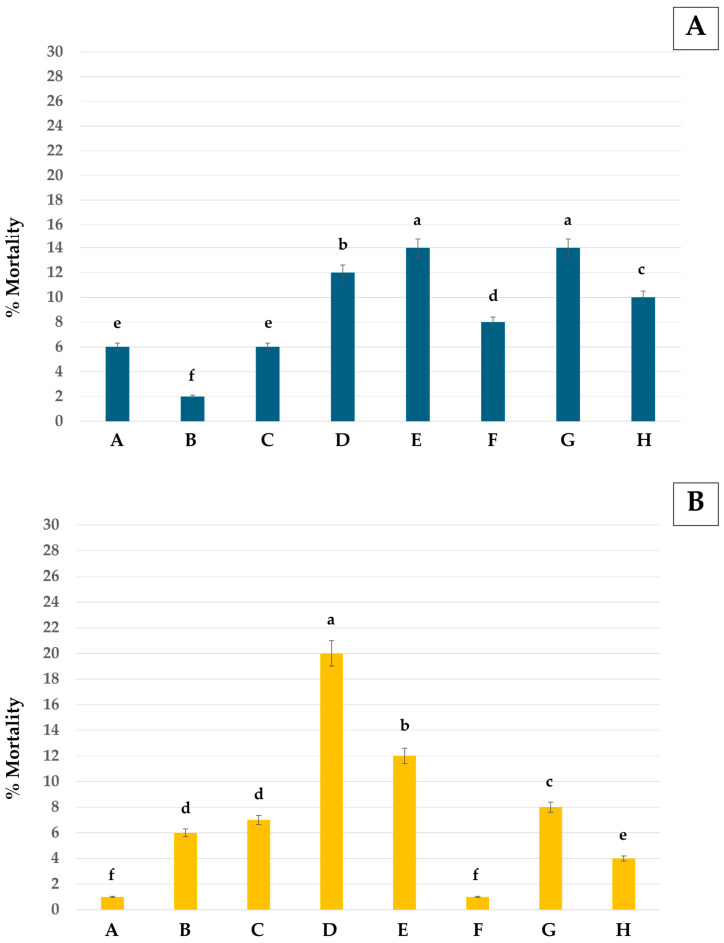
Mortality rate against non-target species of pure monoterpenes and mixture formulations: (**A**) guppies and (**B**) honeybees. Standard errors are demonstrated as error bars. Note: Values that are accompanied by different letters (a–f) indicate significant differences between the formulations. **A**, eucalyptol; **B**, geranial; **C**, *trans*-anethole; **D**, *trans*-cinnamaldehyde; **E**, eucalyptol + *trans*-cinnamaldehyde; **F**, eucalyptol + geranial; **G**, *trans*-anethole + eucalyptol; and **H**, *trans*-anethole + geranial.

**Figure 10 insects-16-00738-f010:**
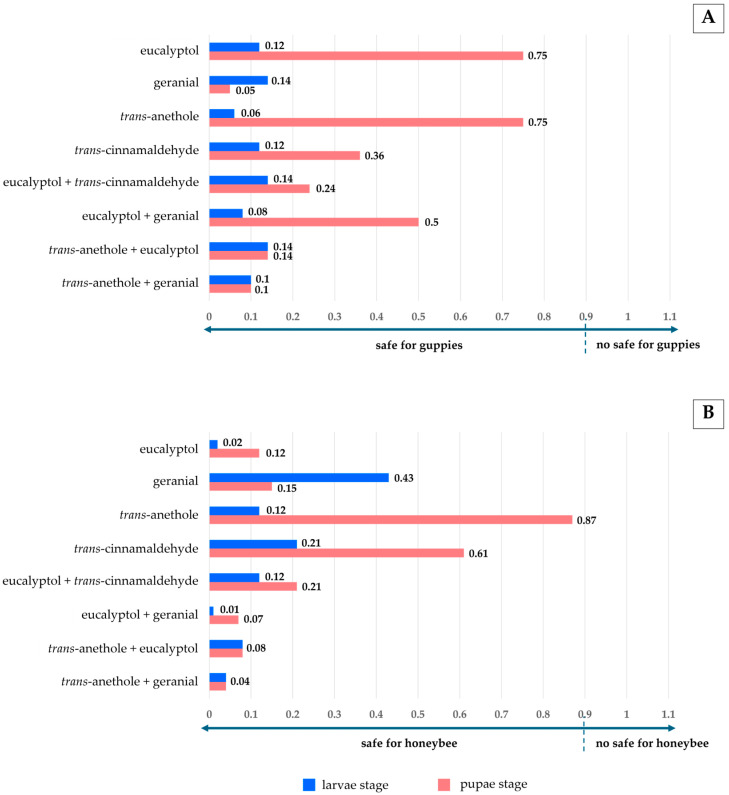
Biosafety index (BI) of pure monoterpenes and mixture formulations against guppies (**A**) and honeybees (**B**). Note: BI is the biosafety index, determined by %mortality of non-target species (honeybees or guppies) divided by %mortality of larvae or pupae of the mosquitoes.

**Table 1 insects-16-00738-t001:** Pure and mixed formulations of monoterpenes against mosquitoes.

Pure and Mixed Forms.	Activity Against	Efficiency	Ref.
Eucalyptol	*Ae*. *aegypti* larvae	LC_50_ = 104.5 µg/mL	[21]
	*Culex quinquefasciatus* larvae and pupae	LC_50_ = 44.4 and 92.9 µg/mL	[22]
Geranial	*Cx*. *quinquefasciatus* larvae and pupae	LC_50_ = 53.4 and 193.9 µg/mL	[22]
Limonene	*Cx*. *quinquefasciatus* larvae and pupae	LC_50_ = 27.3 and 98.4 µg/mL	[22]
Carvacrol	*Ae*. *aegypti* larvae	LC_50_ = 8.8 µg/mL	[23]
Phenyl acetic acid	*Ae*. *aegypti* larvae	LC_50_ = 3.81 ppm	[24]
*Trans*-anethole	*Ae*. *aegypti* larvae	LC_50_ = 88.5 mg L^−1^	[25]
	*Ae*. *aegypti* larvae and pupae	LC_50_ = 2.5% and 3.3%	[16]
*Trans*-cinnamaldehyde	*Anopheles gambiae* larvae	LC_50_ = 0.06 g/L	[26]
Geranial + *trans*-cinnamaldehyde (1:1)	*Ae*. *aegypti* larvae and pupae	LT_50_ = 0.2 h	[17]
*Trans*-anethole + γ-terpinene (1:1)	*Ae*. *aegypti* larvae	LC_50_ = 12.4 ppm	[27]
Methyl cinnamate + linalool (1:1)	*Ae*. *aegypti* larvae	LC_50_ = 57.7 µg/mL	[28]

**Table 2 insects-16-00738-t002:** Estimates of the LC_50_ and LC_90_ of pure monoterpene and mixture against *Ae*. *aegypti* larvae after 24 h of exposure.

Treatment	Lethal Concentration(50% or 90%)	Estimated Concentration (µg/mL)	CI_95_ (µg/mL)	Slope ± SE	ICV
eucalyptol	LC_50_	382	353–416	0.011 ± 0.002	
	LC_90_	496	452–584		
geranial	LC_50_	672	502–1592	0.003 ± 0.002	
	LC_90_	1047	729–2870		
*trans*-anethole	LC_50_	299	271–328	0.021 ± 0.003	
	LC_90_	362	332–401		
*trans*-cinnamaldehyde	LC_50_	267	239–295	0.012 ± 0.002	
	LC_90_	376	341–429		
eucalyptol + *trans*-cinnamaldehyde	LC_50_	189	-	0.023 ± 0.006	0.3
	LC_90_	244	-		
eucalyptol + geranial	LC_50_	219	-	0.022 ± 0.005	0.2
	LC_90_	277	-		
eucalyptol + *trans*-anethole	LC_50_	176	-	0.025 ± 0.005	0.3
	LC_90_	228	-		
*trans*-anethole + geranial	LC_50_	183	-	0.024 ± 0.003	0.2
	LC_90_	236	-		

LC_50,90_ = lethal concentration required to kill 50% and 90%. CI_95_ = 95% confidence intervals; exposure concentration is considered significantly different when the 95% CI fails to overlap. ICV = increased concentration value.

**Table 3 insects-16-00738-t003:** Estimates of the LC_50_ and LC_90_ of pure monoterpene and mixtures against *Ae*. *aegypti* pupae after 72 h of exposure.

Treatment	Lethal Concentration(50% or 90%)	Estimated Concentration (µg/mL)	CI_95_ (µg/mL)	Slope ± SE	ICV
eucalyptol	LC_50_	770	539–5066	0.003 ± 0.000	
	LC_90_	1151	746–9035		
geranial	LC_50_	445	393–552	0.007 ± 0.002	
	LC_90_	642	540–898		
*trans*-anethole	LC_50_	761	-	0.004 ± 0.002	
	LC_90_	1095	-		
*trans*-cinnamaldehyde	LC_50_	470	406–617	0.006 ± 0.001	
	LC_90_	702	574–1042		
eucalyptol + *trans*-cinnamaldehyde	LC_50_	364	327–413	0.007 ± 0.001	0.3
	LC_90_	535	471–656		
eucalyptol + geranial	LC_50_	518	443–815	0.007 ± 0.002	0.4
	LC_90_	714	570–1356		
eucalyptol + *trans*-anethole	LC_50_	252	224–744	0.023 ± 0.011	0.2
	LC_90_	308	256–1355		
*trans*-anethole + geranial	LC_50_	167	-	0.026 ± 0.011	0.1
	LC_90_	217	-		

LC_50,90_ = lethal concentration required to kill 50% and 90%. CI_95_ = 95% confidence intervals, exposure concentration is considered significantly different when the 95% CI fails to overlap. ICV = increased concentration value.

## Data Availability

All relevant data are included in the article.

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
