# Peer review of "Synergistic Larvicidal and Pupicidal Effects of Monoterpene Mixtures Against Aedes aegypti with Low Toxicity to Guppies and Honeybees"

_insects, 2025, doi:10.3390/insects16070738_

Round 1
Reviewer 1 Report
Comments and Suggestions for Authors
This study focuses on the insecticidal activity of mixed monoterpenes against the larvae and pupae of Aedes aegypti, as well as their safety to non-target organisms, providing abundant data. The experiments demonstrated that two monoterpene mixtures (eucalyptol + trans-anethole and trans-anethole + geranial) exhibit significant insecticidal effects on the larvae and pupae of Aedes aegypti, while being safe to non-target organisms. The experimental design is reasonable, and the data are accurate. However, there are still some details that need improvement.
Abstract
It is suggest that emphasizing the significant insecticidal efficacy of two monoterpene blends (eucalyptol + trans-anethole and trans-anethole + geranial) against Aedes aegypti larvae and pupae.
State key LC50 upfront.
Introduction
Clarify why these specific mixtures were chosen over other combinations. Reference prior synergy studies more directly.
Temephos Resistance: Strengthen discussion of temephos resistance in Ae. aegypti (cite recent global reports)
Methods
Explain why 200/400 µg/mL were selected for bioassays. Include preliminary range-finding data or citations.
Specify how LC50/LC90 values were calculated; Clarify why some LC90 values lack CI
Discussion
Emphasize how monoterpene mixtures could delay resistance vs. temephos.
Line 62-67 “Temephos, an organophosphate larvicide, is a widely…” Simplify this sentence.
Line 110 After the first appearance of essential oils (EOs), the following text should uniformly use Eos.
Line 164 “The assay was done five times for each treatment” Was the same batch of larvae used in the experiment?
Line 2504 Replace "spectacularly" with "notably" or "significantly."
Line 379” that all treatments were extremely safe” Is there any data to support this conclusion?
Line 467 “see Figures 6 and 7” Should change to ”Figure 6, 7”.
Improve microscopy image resolution (Figs 7–8); use scale bars.
Reviewer 2 Report
Comments and Suggestions for Authors
The manuscript has several áreas of opportunity, both in substance and form. Authors should consider the order of the statistical analysis in order to write the results and discussion. The graphs and figures require careful analysis, as they are very small and the images should provide a reference point for determining measurenment.
In the toxicology studies, it is normal that some mortality caused by handling of biológicas material ocurrs. It is necessary to view the mortality either in tables or in the graphs that can be included.
When the significant difference is cited in the text, the probablity should be included, as for the ANOVA analysis and means test,it is not clearly described inthe results. About 52% of the literatura is recent.

Reviewer 3 Report
Comments and Suggestions for Authors
See attached document

The treated pupae showed cell damages… should be cell damage.
They were as effective as temephos better phrased as “comparable in efficacy to temephos.
Inconsistent switching between past and present tenses in results and discussion (e.g., showed, demonstrates, implies). Use past tense for results consistently.
trans-anethole + geranial sometimes appears with inconsistent formatting. Use consistent italicization or spacing.
Introduce and limit acronyms judiciously. For instance, BI, MI, IMV, ICV, SRI—consider consolidating in a table or glossary.
